REGISTERED REPORT PROTOCOL

# Barriers to anti-retroviral therapy adherence among adolescents aged 10 to 19 years living with HIV in sub-Saharan Africa: A mixed-methods systematic review protocol

Londiwe D. Hlophe [1,2], Jacques L. Tamuzi [1], Constance Shumba[3], Peter S. Nyasulu [1,4] *

1 Faculty of Medicine and Health Sciences, Division of Epidemiology and Biostatistics, Stellenbosch University, Cape Town, South Africa, 2 Faculty of Health Sciences, Department of Environmental Health Sciences, University of Eswatini, Mbabane, Kingdom of Eswatini, 3 School of Nursing and Midwifery, Aga Khan University, Nairobi, Kenya, 4 Faculty of Health Sciences, Division of Epidemiology and Biostatistics, School of Public Health, University of the Witwatersrand, Johannesburg, South Africa

* pnyasulu@sun.ac.za

## Abstract

### Background

Antiretroviral therapy (ART) adherence is fundamental in achieving viral load suppression and consequently attaining positive health outcomes among people living with HIV. However, ART adherence is sub-optimum among adolescents living with HIV (ALHIV) thus the high AIDS-related mortality even after World Health Organization (WHO) revised HIV treatment eligibility guidelines in 2010, 2013 and 2016. Consolidated trends of barriers to ART adherence among ALHIV aged 10 to 19 years in sub-Saharan countries post each eligibility guidelines revision to date are unknown.

### Methods and analysis

We will conduct comprehensive search of peer-reviewed and grey literature databases publishing observational studies reporting data adherence and barriers to ART among ALHIV on ART. We will further search the reference lists of included studies and other relevant reviews. We will also do a citation search for included studies in the review. We will search in the following databases PubMed, Cochrane Review, Scopus on Excerpta Medica Database (Embase) and Cumulated Index to Nursing and Allied Health Literature (CINAHL). Furthermore WHO, Joint United Nations Programme on HIV/AIDS (UNAIDS) websites, conference proceedings and country reports will be searched to identify relevant literature. Data will be extracted from eligible studies and synthesis will be through categorizing studies by year of study, barriers, and outcomes. Meta-analysis and meta-synthesis will be conducted for quantitative and qualitative data, respectively. Where meta-synthesis is impossible, narrative synthesis will be conducted. We will only include studies conducted between 2010 and 2022 within sub-Saharan Africa countries.

**Data Availability Statement:** All relevant data from this study will be made available upon study completion.

**Funding:** The authors received no specific funding for this work.

**Competing interests:** The authors have declared that no competing interests exist.

**Abbreviations:** AIDS, Acquired immunodeficiency syndrome; ALHIV, adolescents living with HIV; ART, antiretroviral; CD4, cluster of differentiation 4; COREQ, Consolidated Qualitative Study; ENTREQ, Enhancing Transparency in Reporting the synthesis of Qualitative research; GRADE, recommendations assessment development and evaluation; HIV, Human Immunodeficiency Virus; MeSH, Medical Search Headings; MMAT, Mixed Methods Appraisal tool; PICO, participants, intervention, comparator and outcome; PRISMA, Preferred Reporting Items for Systematic Reviews and Meta-Analyses; SSA, sub-Saharan Africa; STROBE, Strengthening the Reporting of Epidemiological Studies; UNAIDS, The Joint United Nations Programme on HIV/AIDS; UTT, universal test and treat; WHO, World Health Organization.

## Discussion

Adherence to ART at a high level is required to achieve adequate viral suppression and improve quality of life in ALHIV. The knowledge of barriers to ART among ALHV may aid in the design of interventions aimed at improving ART adherence.

## Trail registration

Systematic review protocol registration: PROSPERO CRD42021284891.

## 1. Background

Globally, ground breaking milestones have been achieved since the first case of Human Immunodeficiency Virus (HIV) was reported in 1981 and the first antiretroviral treatment was approved in 1987 [1–3]. Advances have been made in HIV therapy resulting to improved HIV outcome. For instance in the year 2000, globally, over 5 million new cases were reported with 14 000 new cases daily while in 2020, only 1.5 million new infections were reported [1,4]. Furthermore, acquired immunodeficiency syndrome (AIDS) related deaths have decreased by approximately 50% since 2010 [5]. However, the HIV treatment outcomes are noticeable among the general population while adolescents still present negative HIV outcomes [6]. In the year 2020, there were 37.6 million living with HIV of which adolescents aged 10 to 19 years accounted for a 5% of these cases. A total of 73% of the global population living with HIV were on treatment while only 54% adolescents living with HIV (ALHIV) were on treatment. While viral load suppression rate is improving globally, adolescents still report a very low viral load suppression rate [6–11]. Subsequently, there has been an observed reduction in terms of AIDS related mortalities globally as in 2020, there were only 690 000 AIDS related deaths compared to the 2010 where there were 1.2 million deaths. This indicates a 60% reduction in mortality compared to 2004 peak deaths [6,12].

Nevertheless, HIV prevalence among adolescents and young people is projected to reduce by 61% globally and by 84% in the sub-Saharan Africa region between 2010 and 2050 [13]. Secondly, the UNAIDS has set a target of reducing new HIV infections and AIDS-related deaths by 90% in 2030 compared to 2010 through implementing the treatment cascade to achieve viral suppression [14,15]. However, adolescents still present poor HIV outcomes as a result of poor retention in HIV care and poor antiretroviral therapy (ART) adherence which may hinder achievements of these projections and targets [9,13]. For instance, even though AIDS-related deaths have been dropping among the adults, it has increased by 45% among ALHIV between 2005 and 2015 [7,9]. Data from sub-Saharan Africa (SSA) region indicates that the outcomes are even more exaggerated in this region among ALHIV [6,10]. ART adherence is vital in improving the HIV outcomes [16]. Optimum ART adherence is defined as >95% of correctly taking ART medication [15,17].

Nevertheless, enormous efforts have been made to curb the HIV pandemic globally. This includes the revision of HIV treatment eligibility guidelines by WHO aimed at improving HIV outcomes. The first guidelines were published in 2002 where a CD4 less or equal to 200cells/mm$^3$ and or an AIDS-related condition was the recommended threshold to start ART among those living with HIV. These guidelines were further revised and the threshold was increased to CD4 $\leq$350 cells/mm$^3$ including all pregnant women and person with stage 3 or 4 of the WHO HIV infection stages in 2006 [18]. WHO broadened the threshold in 2010 to CD4$\leq$350

cells/mm$^3$ for all regardless of the clinical stage [19]. Furthermore, in 2013, WHO increased the recommended threshold to CD4≤500 cells/mm$^3$ for children above 5 years and adults irrespective of clinical stage or symptoms [20]. The recommendation was adopted by countries and improved the ART coverage. Consequently, in 2015, WHO recommended the universal test and treat (UTT) regardless of clinical stage, CD4 cell count or symptoms. This recommendation was mostly revised by countries in 2017 [21], although countries in SSA delayed adoption of the WHO guidelines due to costs related to providing ART, shortage of staff and drugs in healthcare facilities and the limited training of staff soon after guidelines are changed [22,23].

Although the recommendations have improved the HIV outcomes among those on ART, adolescents still mainly present negative HIV outcomes compared to general population [24–27]. There is therefore a need to establish the barriers faced by adolescents with ART adherence despite the evidence-based guidelines as they have a negative baring on the 2010 to 2050 projections and the UNAIDS 2030 target.

However, data are sparse to support a comprehensive understanding of challenges faced by adolescents that compromise their ability to adhere to ART. We believe that diferentiated focus on adolescents issues would help unravel potential solutions targetted towards adolescents reformed behaviour that would promilgate improved adherence to treatment and improve treatment outcomes. So far most data has been aggregated with adults and not stratified to adolescents making it difficult to elucidate what adolescents focused challenges are [28]. Secondly, available data on adolescents even if stratified by the different age groups, included studies conducted before the 2010 WHO treatment guidelines whereby treatment elligibility was based on the WHO HIV infection stages [29]. Lastly, other studies classify adolescence as aged 12 to 24-year-old which include young people as per the WHO definition of adolescents and young prople [30].

Therefore this review seeks to focus on adolescents aged 10 to 19 years as per WHO definition of adolescents and secondly studies conducted within the projections and post the 2010 treatment guidelines. The criteria for including studies conducted post the 2010 treatment guidelines is due to evidence that patients starting ART at HIV advanced stage were associated with adverse HIV outcomes [31]. Secondly, patients who were not started on ART because of high CD4 count were associated with low retention to care and associated with high mortality and morbidity due to AIDS progression [32,33].Therefore, with the revised eligiability guidelines, which recommended ART initiation regardless of the clinical stage, we believe studies including adoelscents on ART post the 2010 guidelines will therefore present comparable barriers to ART. Secondly, by focusing on adolescents, we believe understanding specifics about adolescents adherence bahaviour is key to improved treatment compliance and outcomes. Such knowledge would support developmet of specific approaches that would enhance adolescents adherence to ART and ultimately lead to an improvement in treatment outcome

## 2. Methods and design

This study protocol was registered with the International Prospective Register of Systematic Reviews, PROSPERO (CRD42021284891). The results will be reported according to the Preferred Reporting Items for Systematic Reviews and Meta-Analyses (PRISMA statement) [34], and the Enhancing Transparency in Reporting the synthesis of Qualitative research (ENTREQ) recommendations [35].

The objective of this mixed method systematic review will be to identify the barriers to ART adherence among ALHIV aged from 10 to 19 years in sub-Saharan Africa post-2010 treatment guidelines to 2022.

## 2.1. Eligibility criteria

**2.1.1. Inclusion criteria.** Study design: We will include quantitative, qualitative, and mixed studies in the review. Quantitative studies will include cross-sectional, case-control studies and cohort studies and qualitative studies will include interviews, focus groups, ethnographic observations, phenomenological, participatory action research, and surveys. Mixed studies include both quantitative and qualitative designs. Only studies published in English will be considered in this review.

Study participants: The study participants for this review will include studies conducted on adolescents aged 10 to 19 years living with HIV and on ART. We also will consider studies including other age group only if they disintegrate the study population by age group in their analysis.

Interventions: The review will be for observational studies reporting on level of ART adherence and barriers to ART among adolescents living with HIV.

Outcomes: We will include only studies reporting on ART outcomes among adolescents (10 to 19 years) and the following will be the outcomes of interest in this review:

- ART adherence rate: proportion of adolescents with optimum ART adherence among total adolescent in the study

- ART adherence barriers: list and proportion of adolescents reporting either patient or individual, economic, social, health system, therapy-related and cultural based barriers faced by participants in attaining optimum ART adherence [36–39].

- Lost to follow up (LTFU): proportion adolescents lost to follow up.

- Viral load (VL) suppression: proportion of adolescents acquiring VL suppression (VL < 40 copies/μL) in the study.

- Unsuppressed VL: proportion of adolescents with VL>40 copies/μL.

- CD4 T cell count: proportion of adolescents reporting CD4 T count as per WHO clinical stages.

Time and setting: The review will include studies conducted between 2010 and 2022 in SSA countries as per World Bank classification.

**2.1.2. Exclusion criteria.** The review will exclude the following studies:

- Studies conducted outside the SSA

- Studies conducted on other age groups outside adolescents or have not disaggregated findings as per the different age groups.

- Studies reporting on pre-exposure prophylaxis or post-exposure prophylaxis adherence among adolescents

- Studies in other languages except English

- Studies presenting abstracts only

- Mathematic model studies, opinion pieces, editorials, qualitative reviews, and quantitative reviews

## 2.2. Search strategy

We will develop a sensitive search strategy using participants, intervention, comparator, and outcome (PICO) keywords for the search terms. We will develop a comprehensive set of terms

for the search strategy using PubMed Medical Search Headings (MeSH) and list of synonyms of the terms using our expertise of the field. We will use PubMed to develop the search strategy and this search strategy will be adapted for other electronic databases. The review will use PubMed, Cochrane Review, Scopus on Excerpta Medica Database (Embase) and CINAHL for electronic biographic databases search of studies. Reference lists of included studies will also be assessed for more relevant studies and reviews. We will also conduct a citation search for included studies and relevant reviews for more eligible studies.

Additionally, we will search for adolescents ART adherence reports from World Health Organization (WHO) and Joint United Nations Programme on HIV/AIDS(UNAIDS) websites. We will also include reports from the International Conference on HIV/AIDS, International AIDS Society, and International Conference on AIDS and STIs in Africa on ART adherence among adolescents. Furthermore, we will search for unpublished country reports; with specific emphasis on countries reporting eminent poor ART adherence and or poor ART outcome among adolescents in sub-Saharan Africa.

The following search terms will be used to select studies: anti-retroviral therapy or ART adherence, barriers to anti-retroviral therapy, viral load suppression or CD4 T count, adolescents, and sub-Saharan Africa or SSA. Studies will be filtered by year of publication thus only including studies conducted between 2010 (when WHO first extended HIV treatment eligibility to individuals with less advanced disease) to date, 2022. A detailed summary search terms that will be used in combination in the search strategy is presented as supporting information below (S1 Table). We will manage the search using Mendeley software and this will include importing studies to Mendeley desktop library and removal of duplicates. We will prepare a standard report template including the databases searched, keywords used and number of search results. Retrieved studies will be saved for further assessment.

## 2.3. Quality and eligibility assessment

Full text articles with relevant titles and abstracts will be evaluated for quality and eligibility by two investigators independently. Discrepancies will be discussed and resolved by both investigators and in consultation with a third investigator. Quality assessment will be ensured through having a document and report-search process guided by the PRISMA-P 2015 checklist (S2 Table) [34] and the Enhancing Transparency in Reporting the synthesis of Qualitative research (ENTREQ) recommendations [40]. The study selection outline will be summarized using the modified PRISMA flow-chart as indicated figure in S1 Fig. We will develop a table to indicate both included and excluded studies based on the review eligibility criteria with motivation for excluded studies.

## 2.4. Data extraction and management

Two investigators will develop a list of sub-topics to be used to extract information from each study. The list will be pre-tested by cross-checking, what needs to be extracted before the actual. During this process, even though independently conducted, frequent communication will be maintained between the two investigators. In a case of a missing variable in a study, we will contact the corresponding author of the study through an email. However, where we are unsuccessful, the missing data and its implication in the overall review will be reported. Following agreement on qualified studies, authors will enter relevant data from potentially eligible studies into the excel data extraction forms. To guide the design of these types, the Strengthening the Reporting of Epidemiological Studies (STROBE) and the Consolidated Qualitative Study (COREQ) Reporting Standards for both quantitative and qualitative studies will be used [41].

Information on the following variables will be extracted from each study included: the author, the published year, the setting of studies, focus/aim of study, study designs, study population, intervention strategy, barriers to ART adherence faced by study population and outcome measurements. Missing data within categories will be requested from authors through an email. All interviews will be read repeatedly by two members of the research team, and any mention of adolescents will be coded under an overall theme regarding participant–adolescent interaction; the interviews will be further analysed and agreed upon with a third coder. All sub-themes concerning participant–adolescent interaction will be compared within each interview in terms of broader HIV experiences and across interviews until no new insights emerge.

## 2.5. Assessment of risk of bias

We will use the "Mixed Methods Appraisal tool" (MMAT), version 2018 [42], due to the methodological diversity of the studies that will be included in the review. The MMAT specifically addresses items related to the various study designs being evaluated in a single tool, thus the MMAT is a critical appraisal tool designed for the appraisal process of mixed methods reviews [42]. The MMAT has methodological quality criteria for (a) qualitative research, (b) randomized controlled trials, (c) non-randomized studies, (d) quantitative descriptive studies, and (e) mixed methods studies. This tool has been widely used in systematic reviews, according to the Cochrane Qualitative and Implementation Methods Group, and has the advantage of being able to assess interdependent qualitative and quantitative elements of mixed-methods research [42]. Using the MMAT, we will independently assess methodological quality based on the details published in the studies that will be included in our review. Any disagreements will be resolved through dialogue. We will document all individual ratings for each study included in this review to ensure the transparency of the risk of bias assessment. We will discuss whether to exclude studies that fail to meet more than one quality criterion.

## 2.6. Data analysis and synthesis

For quantitative data, where possible, risk ratio or odds ratio for the following categorical outcomes will be computed. We will also plot the proportions of ART adherence rate, ART adherence barriers, viral load suppression and unsuppressed, and CD4 count with their 95% confidence intervals will be calculated from the data generated by each included study. If appropriate with available data, results from comparable groups of studies will be pooled into statistical meta-analysis using Stata software version 16. The Quantum Geographic Information System (QGIS) software version 3.24.0 will be used to map included studies across sub-Saharan Africa. Effect size data will be analyzed using a random-effects model, and a combined effect estimate will be computed through a meta-analysis. In addition to this, summary tables characterizing studies and results will be constructed for the meta-analysis of quantitative studies. Heterogeneity between combined studies will be tested using standard chi-square test. We will use the $I^2$ to describe the percentage variation of included studies [40]. We anticipate heterogeneity due to clinical (as a result of treatment model adopted by countries to facilitate ART adherence among adolescents) and methodological (due to methods used in the different studies such as subject selection and allocation criteria) variation thus we will conduct subgroup analysis such as subset of type of barriers and countries. Furthermore, meta-regression will be performed to examine the association between different adolescents age groups and ART adherence. Where statistical pooling is not possible the findings will be presented in narrative form. Besides, publication bias will be assessed in quantitative studies included in the systematic review and meta-analysis; we will conduct sensitivity analysis and plot results against the fit of a funnel plot to visually assess publication bias asymmetry. Furthermore, the

Egger and Begg & Mazumdar rank correlation tests will be used to test for the funnel plot asymmetry.

For qualitative data, where meta-synthesis is possible, textual data will be pooled using the Qualitative Assessment and Review Instrument (JBI-QARI) and Narrative, Opinion and Text Assessment and Review Instrument (JBI-NOTARI) package from the SUMARI software suite. Data will be analysed using framework methods, using both a case- and theme-based approach. Where textual pooling is not possible the findings will be presented in narrative form [43].

### 2.7. Assessment of quality of evidence

We will use grading of recommendations assessment development and evaluation (GRADE) software to grade for quality of evidence of the review. We will consider the five criteria: risk of bias, heterogeneity, indirectness, impression, and publication bias. We will present the findings of our grading in a summary table where level of certainty will be graded as high, moderate, low, and very low.

## 3. Discussion

The study will include quantitative, qualitative, and mixed study designs thus allowing for a broad analysis and conclusion. This review will only include studies conducted in sub-Saharan countries and therefore evidence generated will be representative of ART adherence among adolescents in the region allowing for evidence based future programming and studies. In our knowledge, this review will be the first study to track the barriers to ART adherence among ALHIV aged 10 to 19 years in Sub-Saharan Africa after the WHO recommendations of 2010 in relation to the 2010 to 2050 projections.

In fact, a high level of adherence to ART is required to achieve adequate viral suppression and improve quality of life in ALHIV. Knowledge of this specific group's barriers such as ALHIV may aid in the design of interventions aimed at improving adherence to ART. However, the consolidated trends of barriers to ART adherence among ALHIV aged 10 to 19 years in Sub-Saharan Africa following each eligibility guidelines revision are unknown. This systematic review is set following the three WHO recommendations post-2006, this is substantial to better understand the experiences of adolescents on poor and successful ART adherence. Knowing that ART adherence may include multifactorial with multifaceted individual, family/caregiver, and hospital barriers, the transitional aged of adolescence need more attention in ALHIV as the success of young adulthood HIV treatment cascade depend on the previous age.

However, since the review will only include studies conducted in English, there might be selective bias thus findings of the study might not be entirely conclusive.

## 4. Conclusion

The study aims to ascertain ART adherence among adolescents in SSA, barriers to ART adherence and ART outcome and trend of these between 2010 and 2022. The review will be conducted in multiple phases, studies' searching, studies' screening, data extraction and analysis. The results of the review will inform the broader study's conceptual framework development for intervention aimed at improving ART adherence through addressing barriers to ART adherence among adolescents in the SSA.

## Supporting information

**S1 Fig. Modified PRISMA flow-chart.**
(TIF)

**S1 Table. Search strategy for the systematic literature review.**
(TIF)

**S2 Table. PRISMA-P 2015 checklist.**
(TIF)

## Author Contributions

**Conceptualization:** Londiwe D. Hlophe, Peter S. Nyasulu.

**Writing – original draft:** Londiwe D. Hlophe.

**Writing – review & editing:** Jacques L. Tamuzi, Constance Shumba, Peter S. Nyasulu.

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
