## [Decision Letter · Decision Letter 0]

27 Mar 2022

PONE-D-21-39720Barriers to anti-retroviral therapy adherence among adolescents aged 10 to 19 years living with HIV in sub-Saharan Africa: A mixed-methods systematic review protocolPLOS ONE

Dear Dr. Nyasulu,

Thank you for submitting your manuscript to PLOS ONE. After careful consideration, we feel that it has merit but does not fully meet PLOS ONE’s publication criteria as it currently stands. Therefore, we invite you to submit a revised version of the manuscript that addresses the points raised during the review process.

We look forward to receiving your revised manuscript.

Kind regards,

Professor Kwasi Torpey, MD PhD MPH

Academic Editor

PLOS ONE

Journal Requirements:

4. We note that you have referenced (ie. Bewick et al. [5]) which has currently not yet been accepted for publication. Please remove this from your References and amend this to state in the body of your manuscript: (ie “Bewick et al. [Unpublished]”) as detailed online in our guide for authors

Additional Editor Comments (if provided):

This study is extremely important as it seeks to highlight our understanding of adherence among adolescents. The manuscript is nicely written and very easy to follow. In East and south Africa region 90% of PLHIV know their status 78% are on ART and 72% are virally suppressed. for West Africa it is 81%-77%-62%. Whilst the authors have given some data, highlighting the proportions among adolescents compared to adults makes a compelling case. In addition, the authors should correct minor typo in the methods section (pinion piece - opinion piece)

Reviewers' comments:

Reviewer's Responses to Questions

**Comments to the Author**

1. Does the manuscript provide a valid rationale for the proposed study, with clearly identified and justified research questions?

Reviewer #1: Yes

2. Is the protocol technically sound and planned in a manner that will lead to a meaningful outcome and allow testing the stated hypotheses?

Reviewer #1: Yes

3. Is the methodology feasible and described in sufficient detail to allow the work to be replicable?

Reviewer #1: Yes

4. Have the authors described where all data underlying the findings will be made available when the study is complete?

Reviewer #1: No

5. Is the manuscript presented in an intelligible fashion and written in standard English?

Reviewer #1: Yes

6. Review Comments to the Author

You may also provide optional suggestions and comments to authors that they might find helpful in planning their study.

Reviewer #1: The manuscript presents the protocol of a systematic review for barriers of adherence to ARVs among HIV positive adolescents in SSA. Adherence to ARVs has been reported to be lower in adolescents and this has has also been reflected in their negative treatment outcomes. This study hence seeks to understand the barriers of adherence to ARVs among adolescents in a systematic review study. The authors have chosen to review studies from 2010 when WHO reviewed its treatment guidelines.

The manuscript is well written and structured. The authors have identified the problem and the research gap. It is also novel as it is a mixed method systematic review. The methods for the review for both quantitative and qualitative studies have been well explained in the methods section.

The manuscript does not have any major weakness. However, in terms of relevance, the authors indicate that the study will be the first to track barriers of ART but it is not clear how this information will be useful in changing the negative HIV treatment outcomes in adolescents. It is not clear what new knowledge this study adds to existing studies that have reported barriers of adherence to ARVs among adolescents. I recommend that the authors could consider the use of the mixed method approach to show the added value this study will have to previous research.

7. PLOS authors have the option to publish the peer review history of their article (what does this mean?). If published, this will include your full peer review and any attached files.

Reviewer #1: No

---

## [Author Response · Author response to Decision Letter 0]

18 Apr 2022

Dear reviewer,

We are very thankful for your valuable comments for the improvement of our manuscript entitled: “Barriers to anti-retroviral therapy adherence among adolescents aged 10 to 19 years living with HIV in sub-Saharan Africa: A mixed-methods systematic review protocol”. We have tried to address all the comments which are reflected below.

Comment 1: Please ensure that your manuscript meets PLOS ONE's style requirements, including those for file naming.

Response 1: Thanks. The authors have thoroughly edited the manuscript and are assured that it is now well aligned with the journal’s requirements. We have also edited the references into squared brackets as per the journal requirements.

Comment 2: PLOS requires an ORCID ID for the corresponding author in Editorial Manager on papers submitted after December 6th, 2016

Response 2: Thanks. The ORCID ID (0000-0003-2757-0663) for the corresponding author has been added.

Comment 3: We note that you have indicated that data from this study are available upon request. 

Response 3: Amendment done. The authors have revised this sentence accordingly to state that data will be available to the scientific community without restrictions for replication of study. A supporting information file of the search strategy has been attached to this manuscript (S1 Table1). On production of the main manuscript, data will be attached as s supporting information for access by anyone interested in the data.

Comment 4: We note that you have referenced (ie. Bewick et al. [5]) which has currently not yet been accepted for publication.

Response 4: Many thanks. Bewick et. al was not referenced in the study instead reference [5] is UNAIDS. Global HIV & AIDS statistics — Fact sheet , 2021, retrieved from: https://www.unaids.org/en/resources/fact-sheet (Line 70 and 361) 

Comment 5: Additional Editor Comments (if provided): This study is extremely important as it seeks to highlight our understanding of adherence among adolescents. The manuscript is nicely written and very easy to follow. In East and south Africa region 90% of PLHIV know their status 78% are on ART and 72% are virally suppressed. for West Africa it is 81%-77%-62%. Whilst the authors have given some data, highlighting the proportions among adolescents compared to adults makes a compelling case. In addition, the authors should correct minor typo in the methods section (pinion piece - opinion piece)

Response 5: Thanks for this obeservation. The following typos have been edited including any others throught out the text. Line 5: “livng” has been corrected to “living” and in the line 172: “pinion pieces” has been corected to “opinion pieces”.

Reviewer #1

Comment 1: The reviwer noted that authors have declared that data will be available on request.

Response 1: The authors have revised and stated clearly that data of this study will be available without any restrictions. A supporting information file, S1 Table2: Search strategy for the systematic literature review has been attached. When the full manuscriot is compiled, we will include data as supporting information (Line 199).

Comment 2: The manuscript presents the protocol of a systematic review for barriers of adherence to ARVs among HIV positive adolescents in SSA. Adherence to ARVs has been reported to be lower in adolescents and this has also been reflected in their negative treatment outcomes. This study hence seeks to understand the barriers of adherence to ARVs among adolescents in a systematic review study. The authors have chosen to review studies from 2010 when WHO reviewed its treatment guidelines.

The manuscript is well written and structured. The authors have identified the problem and the research gap. It is also novel as it is a mixed method systematic review. The methods for the review for both quantitative and qualitative studies have been well explained in the methods section. The manuscript does not have any major weakness.

Response 2: Thank you for the positive feedback.

Comment 3: However, in terms of relevance, the authors indicate that the study will be the first to track barriers of ART but it is not clear how this information will be useful in changing the negative HIV treatment outcomes in adolescents. It is not clear what new knowledge this study adds to existing studies that have reported barriers of adherence to ARVs among adolescents. 

Response 3: Data are sparse to support a comprehensive understanding of challanges faced by adolescents that compromise their ability to adhere to ART. We believe that diferentiated focus on adolescents issues would help unravel potential solutions targetted towasrds adolescents reformed behvaiour that would promilgate improved adherence to treatment and improve treatment outcomes. So far, data has been aggregated with adults and not stratified to adolescents making it difficult to elucidate what adolescents focused challenges are. We believe understanding specifics about adolescents adherence bahaviour is key to improved treatment compliance and outcomes. Such knowledge would support development of specific approaches that would enhance adolescents adherence to ART and ultimately lead to an improvement in treatment outcome. (Line 109-120)

Comment 4: I recommend that the authors could consider the use of the mixed method approach to show the added value this study will have to previous research.

Response 4: Many thanks for reminding us to add this point. In page 6, line 124 and 131, we include the following: “The objective of this mixed method systematic review….” And “Study design: We will include quantitative, qualitative and mixed studies in the review”.

---

## [Decision Letter · Decision Letter 1]

2 Jun 2022

PONE-D-21-39720R1Barriers to anti-retroviral therapy adherence among adolescents aged 10 to 19 years living with HIV in sub-Saharan Africa: A mixed-methods systematic review protocolPLOS ONE

Dear Dr. Nyasulu,

Thank you for submitting your manuscript to PLOS ONE. After careful consideration, we feel that it has merit but does not fully meet PLOS ONE’s publication criteria as it currently stands. Therefore, we invite you to submit a revised version of the manuscript that addresses the points raised during the review process.

 At this stage the staff editors from PLOS ONE have some minor concerns regarding your submission which must be addressed for this submission to meet PLOS ONE's publication criteria. These are appended below in the section titled '**Journal Requirements**'. Could you please carefully revise the manuscript to respond to the concerns raised?

We look forward to receiving your revised manuscript.

Kind regards,

Sebastian Shepherd

Staff Editor

PLOS ONE

**Journal Requirements:**

1) In your manuscript you indicate you will include publications from 2010 - 2021. Could you please 

A) Outline why studies published before 2010 will not be included

and

B) Update your manuscript to indicate that you will include studies published in 2022 (to keep the submissions included up to date)

2) In lines 294-296 the rationale for this systematic review is given with the following text: *"In our knowledge, this review will be the first study to track the barriers to ART adherence among ALHIV aged 10 to 19 years in Sub-Saharan Africa after the WHO recommendations post 2006."*

However we have identified the following systematic reviews which discuss barriers to ART adherence:  

Shubber, Z., Mills, E.J., Nachega, J.B., Vreeman, R., Freitas, M., Bock, P., Nsanzimana, S., Penazzato, M., Appolo, T., Doherty, M. and Ford, N., 2016. Patient-reported barriers to adherence to antiretroviral therapy: a systematic review and meta-analysis. PLoS medicine, 13(11), p.e1002183. https://journals.plos.org/plosmedicine/article?id=10.1371/journal.pmed.1002183Kim, S.H., Gerver, S.M., Fidler, S. and Ward, H., 2014. Adherence to antiretroviral therapy in adolescents living with HIV: systematic review and meta-analysis. AIDS (London, England), 28(13), p.1945 https://journals.lww.com/aidsonline/fulltext/2014/08240/adherence_to_antiretroviral_therapy_in_adolescents.12.aspxAmmon, N., Mason, S. and Corkery, J.M., 2018. Factors impacting antiretroviral therapy adherence among human immunodeficiency virus–positive adolescents in Sub-Saharan Africa: a systematic review. Public health, 157, pp.20-31. https://www.sciencedirect.com/science/article/abs/pii/S0033350617304237?via%3Dihub

We are aware that these may differ slightly in population, and that the most recent of these reviews is from 4+ years ago, however, can you please cite these systematic reviews and outline how your work will advance on these works. 

Reviewers' comments:

Reviewer's Responses to Questions

**Comments to the Author**

1. Does the manuscript provide a valid rationale for the proposed study, with clearly identified and justified research questions?

Reviewer #1: Yes

2. Is the protocol technically sound and planned in a manner that will lead to a meaningful outcome and allow testing the stated hypotheses?

Reviewer #1: Yes

3. Is the methodology feasible and described in sufficient detail to allow the work to be replicable?

Reviewer #1: Yes

4. Have the authors described where all data underlying the findings will be made available when the study is complete?

Reviewer #1: Yes

5. Is the manuscript presented in an intelligible fashion and written in standard English?

Reviewer #1: Yes

6. Review Comments to the Author

You may also provide optional suggestions and comments to authors that they might find helpful in planning their study.

Reviewer #1: The authors have shown the relevance of the study and how the findings will be useful in changing the negative treatment outcomes among adolscents

7. PLOS authors have the option to publish the peer review history of their article (what does this mean?). If published, this will include your full peer review and any attached files.

Reviewer #1: No

---

## [Author Response · Author response to Decision Letter 1]

22 Jul 2022

Responses to Reviewers Comments 7th July, 2022

Dear Editor,

We are very thankful for your valuable comments for the improvement of our manuscript entitled: “Barriers to antiretroviral therapy adherence among adolescents aged 10 to 19 years living with HIV in sub-Saharan Africa: A mixed-methods systematic review protocol”. We have tried to address all the comments which are reflected below.

Comment 1.

 In your manuscript you indicate you will include publications from 2010 - 2021. Could you please 

A) Outline why studies published before 2010 will not be included

Response 1A: 

Thanks, the rationale has been explained in line 128 to 136 as follows: “Therefore this review seeks to focus on adolescents aged 10 to 19 years as per WHO definition of adolescents and secondly studies conducted within the projections and post the 2010 treatment guidelines. The criteria for including studies conducted post the 2010 treatment guidelines is due to evidence that patients starting ART at HIV advanced stage were associated with adverse HIV outcomes [31]. In addition to this, patients who were not started on ART due to a high CD4 count had poor retention in care and associated with high mortality and morbidity due to AIDS progression [32, 33]. As a result, we believe that with the revised eligibility guidelines which recommended ART initiation regardless of the clinical stage, we believe studies including adolescents on ART post the 2010 guidelines patients will present comparable barriers to ART." 

B) Update your manuscript to indicate that you will include studies published in 2022 (to keep the submissions included up to date)

Response 1B: Thanks for this valuable input. We have updated the manuscript accordingly. The revised information is on line 40, 151, 182, 219 and 341.

Comment 2. In lines 294-296 the rationale for this systematic review is given with the following text: "In our knowledge, this review will be the first study to track the barriers to ART adherence among ALHIV aged 10 to 19 years in Sub-Saharan Africa after the WHO recommendations post 2006."

However, we have identified the following systematic reviews which discuss barriers to ART adherence: 

• Shubber, Z., Mills, E.J., Nachega, J.B., Vreeman, R., Freitas, M., Bock, P., Nsanzimana, S., Penazzato, M., Appolo, T., Doherty, M. and Ford, N., 2016. Patient-reported barriers to adherence to antiretroviral therapy: a systematic review and meta-analysis. PLoS medicine, 13(11), p.e1002183. https://journals.plos.org/plosmedicine/article?id=10.1371/journal.pmed.1002183

• Kim, S.H., Gerver, S.M., Fidler, S. and Ward, H., 2014. Adherence to antiretroviral therapy in adolescents living with HIV: systematic review and meta-analysis. AIDS (London, England), 28(13), p.1945 https://journals.lww.com/aidsonline/fulltext/2014/08240/adherence_to_antiretroviral_therapy_in_adolescents.12.aspx

• Ammon, N., Mason, S. and Corkery, J.M., 2018. Factors impacting antiretroviral therapy adherence among human immunodeficiency virus–positive adolescents in Sub-Saharan Africa: a systematic review. Public health, 157, pp.20-31. https://www.sciencedirect.com/science/article/abs/pii/S0033350617304237?via%3Dihub

We are aware that these may differ slightly in population, and that the most recent of these reviews is from 4+ years ago, however, can you please cite these systematic reviews and outline how your work will advance on these works. 

Response 2: Thanks for this valuable comment. In the tracking manuscript line 120-136, we have included a paragraph citing these studies. It reads: “So far most data has been aggregated with adults and not stratified to adolescents making it difficult to elucidate what adolescents focused challenges are [28]. Secondly, available data on adolescents even if stratified by the different age groups, included studies conducted before the 2010 WHO treatment guidelines whereby treatment elligibility was based on the WHO HIV infection stages [29]. Lastly, other studies classify adolescence as aged 12 to 24-year-old which include young people as per the WHO definition of adolescents and young prople [30] . Therefore this review seeks to focus on adolescents aged 10 to 19 years as per WHO definition of adolescents and secondly studies conducted within the projections and post the 2010 treatment guidelines. The criteria for including studies conducted post the 2010 treatment guidelines is due to evidence that patients starting ART at HIV advanced stage were associated with adverse HIV outcomes [31].In addition to this, patients who were not started on ART due to a high CD4 count had poor retention in care as well as high mortality and morbidity [32, 33]. As a result, we believe that with the revised eligibility guidelines which recommended ART initiation regardless of the clinical stage, we believe studies including adoelscents on ART post the 2010 guidelines patients will present comparable barriers to ART." 

Comment 3. Please review your reference list to ensure that it is complete and correct. If you have cited papers that have been retracted, please include the rationale for doing so in the manuscript text, or remove these references and replace them with relevant current references. Any changes to the reference list should be mentioned in the rebuttal letter that accompanies your revised manuscript. If you need to cite a retracted article, indicate the article’s retracted status in the References list and also include a citation and full reference for the retraction notice.

Response 3: This has been noted with thanks. In the tracking manuscript, line 460-464, we have revised the following reference: “”WHO 2013. Global update on HIV treatment 2013: results, impact and opportunities: WHO report in partnership with UNICEF and UNAIDS. [Internet]. WHO Press. 2013 [cited 2022 Jun 17]. Available from: https://apps.who.int/iris/bitstream/handle/10665/85326/9789241505734_eng.pdf;jsessionid=22909B1CD56BE4CD2DAF2B462D1D0BC2?sequence=1”

---

## [Editor Report · Decision Letter 2]

9 Aug 2022

Barriers to anti-retroviral therapy adherence among adolescents aged 10 to 19 years living with HIV in sub-Saharan Africa: A mixed-methods systematic review protocol

PONE-D-21-39720R2

Dear Dr. Nyasulu,

We’re pleased to inform you that your manuscript has been judged scientifically suitable for publication and will be formally accepted for publication once it meets all outstanding technical requirements.

Kind regards,

George Vousden

Staff Editor

PLOS ONE
---

## [Editor Report · Acceptance letter]

16 Aug 2022

PONE-D-21-39720R2 

Barriers to anti-retroviral therapy adherence among adolescents aged 10 to 19 years living with HIV in sub-Saharan Africa: A mixed-methods systematic review protocol 

Dear Dr. Nyasulu:

I'm pleased to inform you that your manuscript has been deemed suitable for publication in PLOS ONE. Congratulations! Your manuscript is now with our production department. 

Kind regards, 

on behalf of

Dr. George Vousden 

Staff Editor

PLOS ONE